# Non-invasive Electrolyte Estimation Using Multi-lead ECG data via Semi-supervised Contrastive Learning with an Adaptive Loss

Zhale Nowroozilarki*, Sicong Huang*, Rohan Khera[†], and Bobak J. Mortazavi*

*Department of Computer Science & Engineering, Texas A&M University,
Email: {zhale, siconghuang, bobakm}@tamu.edu
[†]Department of Internal Medicine, Yale School of Medicine, Email: rohan.khera@yale.edu

*Abstract*—**Abnormal electrolyte concentration in blood can lead to serious conditions such as renal failure, high blood pressure, and a life-threatening type of arrhythmia. Electrocardiogram (ECG) is a non-invasive measure of cardiac electrical activity and can capture subtle changes in electrolyte changes. While classification of ECGs with abnormal electrolyte ranges such as potassium levels has been reported, there is a need to leverage continuous ECGs for regressing electrolyte values. Label scarcity is a major limitation in training machine learning-based models for this purpose. Additionally, measured electrolyte values that are accessible in electronic health records datasets typically exhibit a distribution with fewer data points at the extremes which are crucial for accurate prediction. We tackle the above challenges with a pretrained ECG encoder with large but unlabeled datasets to maximize generalization and an adaptive loss that inversely regularizes backpropagation with label frequency. This results in a reduction of 0.1, 0.16, and 0.41 in MAE of the blood potassium, calcium, and sodium level regression tasks respectively, and an improvement of more than 0.15 improvement in the AUROC for the abnormal electrolyte classification tasks when compared against the state-of-the-art models.**

*Index Terms*—**Contrastive Learning, Semi-supervised Learning, ECG, Potassium, Calcium, Sodium, Signal Translation**

## I. INTRODUCTION

**E**LECTROLYTES play a crucial role in biological processes, including the preservation of cellular electrical balance needed in nervous and muscular systems and basic life function [1]. Therefore, accurate estimation of electrolyte levels in the human body is essential in monitoring the individuals' health factors and for improving the quality of care in potentially critical situations [2]. An electrolyte critical to cardiac function is potassium. Abnormal potassium values (dyskalemia), which can be defined as either above normal (hyperkalemia) or below range (hypokalemia), may have significant implications for various severe adverse outcomes such as renal impairment, hypertension, and cardiac arrest [3]. Calcium is another electrolyte whose imbalance range can result in elevated blood pressure and lead to other cardiovascular diseases [4]. Furthermore, an increase in serum calcium level can cause a higher risk associated with Type 2 Diabetes [5]. Sodium is yet another electrolyte critical to function, one that impacts kidneys directly [6]. Therefore timely monitoring and measuring electrolyte levels is crucial for maintaining overall health and preventing potential complications. It ensures that essential bodily functions, such as nerve transmission, muscle contraction, and fluid balance, remain within optimal ranges, thereby supporting normal physiological processes and organ function, preventing adverse health outcomes.

However, obtaining diagnostic electrolyte values requires an ordered lab measured from blood tests, for example, a serum potassium test, which is an invasive method [7]. Invasive procedures often require skilled personnel and specialized equipment, adding to the overall cost, time, and complexity of healthcare delivery. Non-invasive biomarker estimation methods may lead to more patient-friendly, timely, and cost-effective approaches for monitoring key electrolyte levels [8]. In order to monitor the biomarker level in a non-invasive way, there is a need for a data modality that can be used to estimate the fluctuations in the biomarker level. Wearable devices offer non-invasive and user-friendly means of acquiring biosignals which can serve as a proxy in estimating and monitoring patient status, contributing to a shift toward a personalized and accessible healthcare [9]. More specifically, these devices enable continuous monitoring that may potentially enhance patient compliance and allow for early detection and intervention of clinical conditions [10]. Furthermore, biomedical signals provide invaluable information about the health status of various organs.

Variation in the morphology of biomedical waveforms, captured via wearable sensors, can be used as a proxy for the changes in other biomarker levels, as they provide vital information about the physiological and pathological conditions of the human body [11]. As a result, digital biomedical signals can be an invaluable data source for diagnostics, monitoring, research, and the development of personalized and effective healthcare strategies where non-invasive and continuous biomarker measurement is essential.

One of the key biomedical signal modalities that plays a critical role in a variety of assessment, diagnosis, and monitoring tasks is Electrocardiography (ECG) [12]. Abnormality in ECG morphology could indicate electrolyte level imbalances such as calcium or magnesium, and specially with severe ranges of potassium levels [13], [14]. Additionally, monitoring electrolyte levels via ECG may enable potential avenues for noninvasive remote health monitoring, offering greater patient

comfort and seamless health tracking integration without the need for frequent clinical visits [15]. The morphology of a captured ECG signal is indicative of cardiac activity, which may then infer levels of certain biomarkers [16]. Several factors can contribute to changes in ECG morphology from emotional stress to cardiomyopathy and myocardial infarction. These conditions may affect the conduction pathways and electrical activation of the heart, leading to altered ECG patterns [17], [18]. Electrolyte level imbalances such as high or low levels of potassium, calcium, or magnesium, may also impact the electrical activity of the heart and result in changes in ECG morphology prior to such cardiac events [13], [14].

Recent advancements in deep learning models present an opportune avenue for comprehending and analyzing these morphology changes and relating them to underlying changes in electrolyte levels. These models offer the potential to extract insights as a valuable tool in the diagnosis of diverse cardiac conditions through a non-invasive approach [19]. However, deep learning models, particularly neural networks, require a substantial amount of labeled data for models that generalize well, learn complex representations, and perform effectively across diverse and real-world scenario. These labels (ECG data and potassium labs that align in time) are often limited. Acquiring these labels is crucial for training and validating such models, particularly in tasks such as diagnostic predictions. Nevertheless, it is challenging, time-consuming, and costly to collect a large-scale annotated dataset in this field [20]. Consequently, methods based on self-supervised representation learning concepts are gaining popularity in addressing specific challenges related to data scarcity, annotation burdens, and the diverse nature of biomedical datasets [21].

Self-supervised learning methods, such as SimCLR, construct embedding spaces where similar samples are pulled closer while dissimilar data points are pushed further away. This is done by defining each sample as an anchor point and creating an augmentation of the anchor point to serve as the positive pair while treating other samples in the batch as the negative pairs [22]. In other words, each sample in the batch is an anchor point for which we need a positive complement and the other data points in the batch can be treated as negative pairs. As a result, applying these methods requires an augmented version of the data points. Data augmentation in the computer vision domain is a well-researched approach for increasing the quality and the amount of training data [23]. In particular, adding noise or rotating an image does not alter the object type in the image which makes these augmentation techniques suitable.

Applying augmentation to time-series data such as ECG data may introduce a multitude of challenges attributed to the distinctive characteristic inherent in biomedical data related to the underlying physiological feature [24]. First, we need to ensure that we are not adding more noise or unrealistic samples to the training data by applying augmentation. More specifically, it is imperative to ensure that the augmented data retains the distinctive physiological characteristics of the anchor data point without introducing perturbations. Another

challenge in this domain is the lack of access to certain ranges of labeled data which is common in many biomedical datasets. For instance, the number of available ECG samples with a normal range of potassium is larger compared to severe cases.

Therefore, there is a need for a method with an incorporated modality-agnostic augmentation that learns the underlying association between features and labels thoroughly. This work addresses these needs, developing a contrastive learning-based pretraining framework to model ECG data without the need for additional data augmentation, using a customized loss function to estimate blood electrolyte levels from 2 leads of ECG signals to pretrain the feature encoder and solely lead II for the downstream electrolyte regressor.

## II. RELATED WORK

### A. Electrolyte Imbalance Detection

Recent works have shown that an imbalanced range of blood electrolytes can be tracked in ECG tracings. For example, blood potassium levels modify ECG morphology, modifying duration and amplitude of the QRS complex and P and T waves [25]. Additionally, these abnormal blood electrolyte ranges such as hyperkalemia and hypokalemia which refers to the condition of higher or lower concentration of potassium in the blood compared to the normal range, can result in ECG waveforms which are altered. Therefore they can serve as indicators of dyskalemia. Xu et al. derived 12 ECG morphological features to predict hyperkalemia and evaluated five machine learning models on 80 hemodialysis patients, with XGBoost and SVM models achieving the best performance on mild hyperkalemia ($\geq 5mmol/L$) with the XGBoost model and sever cases with the SVM models with area under the receiver operating characteristic curve (AUROC) ranged from 0.740 to 0.931 [26].

Deep neural networks achieved significant success in processing ECG signals for downstream predictions by automatically extracting spatial and temporal features through training [27]. Among the literature, convolutional neural networks (CNN) are the most popular backbone for dyskalemia detection modeling [28]–[30]. Lin et. al. proposed a DenseNet-based ECG12Net for hypokalemia and hyperkalemia detection using 12 leads [29]. While 12-lead ECG can be limited to clinical settings, single-lead ECG data is accessible in wearable and remote health settings. Namely, [30] built a hyperkalemia detection CNN using only 2 leads of ECG. These models do not explicitly address the low-label environments, such as those for dyskalemia.

### B. Electrolyte Concentration Estimation

Estimating potassium concentration values provides more granular details and may lead to a better prognosis than dyskalemia detection, including early identification of changes in potassium values prior to reaching abnormal ranges. Although recent works have demonstrated the feasibility of potassium estimation using single-lead ECG [31], [32], both shared two major limitations of biomedical data: relying on abundant high-quality potassium labels and not accounting for

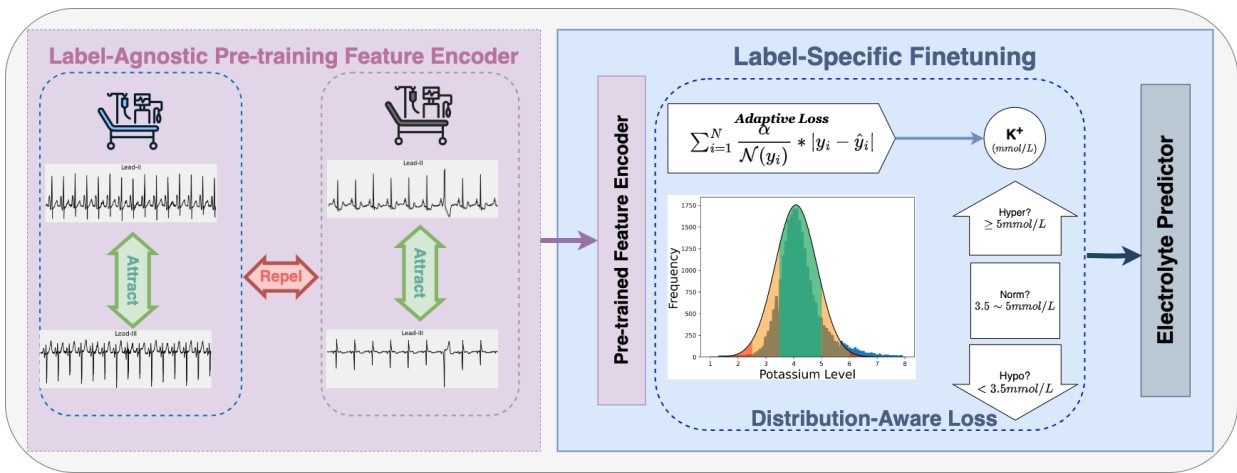

Fig. 1. Our proposed framework starts with self-supervised pretraining to improve generalization using unlabeled ECGs. Instead of augmenting anchors, we use other leads of the same ECG segment to perform contrastive training. During fine-tuning, the contrastive block encodes ECG waveforms into embeddings and passes them to a downstream predictor (i.e. MLP). Predicted values are then used by the proposed adaptive loss by penalizing more on labels with lower occurrences from the dataset. The green region represents the normal range, orange regions represent moderate hyporkalemia and hyperkalem and red regions indicate severe hypokalemia and hyperkalemia.

the skewed distribution of potassium labels between normal vs abnormal ranges.

## III. METHODS

### A. Semi-supervised Contrastive Learning - Pretraining

In this paper we propose ECG2Electrolyte framework in Figure 1, a contrastive-based pretraining coupled with an adaptive loss for fine-tuning using the limited amount of available labeled data, i.e., the electrolyte measurements that occurred within 3 hours before or after the ECG acquisition. The proposed approach learns the underlying representation of ECG morphology by minimizing the distance of two ECG leads of the same data point as they explain the same cardiac activity while maximizing the distance between all other data points from other ICU (Intensive Care Unit) stays. As electrolyte level such as blood potassium level, impacts the ECG morphology, we utilize a convolutional-based encoder with contrastive learning for obtaining an embedding space where similar ECG data points are located in a vicinity. We then employ the aforementioned pertaining schema which constructs an embedding space where similar ECG data points are in a similar region which can be fine-tuned with the specific biomarker value for the downstream task with a minimal amount of labeled data. The representation learning framework comprises two main components:

- Encoder Network: $\mathbf{Enc}(\cdot)$ maps each input $\mathbf{x_i}$ to a corresponding representation $\mathbf{r_i}$ as $\mathbf{r_i} = \mathbf{Enc}(\mathbf{x_i})$ In this work, we use a ResNet50 model as the encoder backbone [33].
- Projector Network: $\mathbf{Proj}(\cdot)$ maps each representation vector $\mathbf{r_i}$ to a corresponding projected vector $\mathbf{z_i}$ using a multi-layer perceptron. The normalized projected vectors allow for computing the similarity score between each pair by performing a dot product between each pair of projected representations: $\mathbf{z_i} = \mathbf{Proj}(\mathbf{r_i})$

As a result, this pretraining process will transform the embedding space where ECG samples with similar morphology will be located in a similar region in the embedding space. Similar to [22], we used a contrastive loss function as follows:

$$\mathcal{L}_{ij} = -\log \left( \frac{\exp((z_i \cdot z_j)/\tau)}{\sum_{k=1}^{2N} \mathbf{I}_{[k \neq i]} \exp((z_i \cdot z_k)/\tau)} \right) \quad (1)$$

Where $I$ constructs a set of all indices of ECG data pairs.

### B. Multi Lead ECG - Augmentation

Contrastive-based representation learning such as SimCLR [22] requires a set of positive and negative pairs whose embeddings are pulled close or pushed apart in the embedding space. However, data augmentation in the medical domain presents unique challenges due to the sensitive and complex nature of medical data. Unlike conventional datasets, medical data and records contain subject-related features, making it challenging to implement traditional augmentation techniques without compromising physiological features.

Furthermore, medical data often involves intricate structures and subtle details crucial for accurate diagnosis and treatment planning. Introducing artificial variations through data augmentation must be carefully executed to avoid distorting critical information that healthcare professionals rely on. Moreover, the scarcity of large and diverse medical datasets adds another layer of complexity, limiting the availability of diverse samples for augmentation. Therefore, the intricate nature of medical data, creates a formidable barrier to the seamless application of data augmentation techniques, requiring innovative approaches and stringent safeguards to ensure data integrity.

Different ECG leads that were acquired at the same time for the same patient are representative of the same cardiac activity [34]. Furthermore, for training a model with a contrastive loss

we need two samples that explain the same event or label to an anchor point [22]. Therefore, instead of applying augmentation to create positive pairs for the contrastive pretraining, in this work, we use ECG lead II data points as anchor points and other available leads as the corresponding positive pairs to that lead. This will eliminate the need for creating an augmented data point for the purpose of positive pair construction while we can treat other patients and ICU stays as negative pairs.

### C. Adaptive Loss Function - Finetuning

The challenges associated with an imbalanced distribution of biomarkers pose significant hurdles in both research and clinical applications. One key challenge is the imbalance data distribution which is common in biomedical datasets, can lead to difficulties in training models that are accurate across the whole possible range of the biomarker. In clinical settings specifically, relying on imbalanced biomarker distributions for diagnostic or prognostic purposes may result in misclassifications, leading to inaccurate assessments of disease risk or progression. Furthermore, addressing these imbalances requires a nuanced understanding of the factors contributing to variations in biomarker levels. Developing robust and standardized measurement techniques becomes crucial to minimize bias and enhance the reliability of biomarker assessments. Ultimately, addressing the challenges associated with imbalanced biomarker distributions is essential for advancing precision medicine and improving the accuracy of diagnostic and prognostic tools.

Similar to many other biomarkers, blood potassium level has a normal distribution where severe-hypokalemia and severe-hyperkalemia samples are minimally represented. To mitigate the issue of imbalanced distribution of potassium and to ensure that the model learns the underlying association between ECG data and any value of potassium, we employ an adaptive loss function. As shown in Figure 2, the electrolyte level follows a distribution where extreme ranges have fewer data points. As a result, we approximate the biomarker value in the population with a normal distribution with the average $\mu$ and standard deviation of $\sigma$ that we obtain from the distribution of the training data. Additionally, each sample is weighted proportional to the inverse of the frequency based on the normal distribution. By applying this modification, we penalize the model more if the prediction is off for the points in the minimally represented regions,

$$Adaptive\ MAE = \sum_{i=i}^{N} \frac{\alpha}{\mathcal{N}(y_i)} * |y_i - \hat{y}_i| \qquad (2)$$

where $y \sim \mathcal{N}(\mu, \sigma^2)$ that approximates the frequency of a particular biomarker value using a normal distribution, $\alpha$ is a tunable coefficient and MAE is Mean Absolute Error.

As a result, the samples from minimally represented regions are penalized proportionally to the corresponding inverse frequency.

The adaptive loss for the classification setting can be written as follows:

$$\mathcal{L} = \alpha * L_1 + \beta * L_2 + \gamma * L_3 \qquad (3)$$

Where $L_1$ is the error for the data points in the batch with potassium level $< 3.5 mmol/L$, i.e., hypokalemia, $L_2$ is the error of the data points in the batch with normal potassium level, and $L_3$ is the error of the data points in the batch with potassium level $\geq 5 mmol/L$, i.e., hyperkalemia. Furthermore, the corresponding coefficient to each range of blood potassium levels is proportional to the inverse frequency of that specific class. This will enable the loss function to apply more weights on the data samples from the range that was represented in the training data distribution the less. As a result, we ensure that the model can learn the underlying association between the ECG morphology and any value of blood potassium level.

Fig 3 illustrates the two-phase processing of training the proposed framework.

## IV. DATA

### A. MIMIC-IV ECG

MIMIC-IV-ECG (Medical Information Mart for Intensive Care IV Diagnostic Electrocardiogram Subset), represents a comprehensive electrocardiogram dataset that can be matched with the MIMIC-IV clinical data [35], [36]. The waveform ECG data captures 12 leads of ECG waveform with a frequency of 500 Hz and the corresponding machine measurements and reports explaining the morphological abnormalities in the data. The MIMIC-IV waveform data used in this work comprised 10-second diagnostic ECG segments. We utilized lead II as the anchor lead and another available lead for the positive pairs from this dataset and sub-sampled them to 100 Hz to reduce the memory need for model training and inference. Detailed demographics of the study cohort for each electrolyte task are provided in Table I.

### B. Blood Potassium Matching to ECG Data

In order to match potassium values as the label for each data, we use the unique Patient ID to find the corresponding clinical data from the clinical dataset as MIMIC-IV clinical data comprise clinical measurements including serum potassium level. Each measurement in this dataset is identified by a unique item ID such as 227442 for blood potassium level. Consequently, we select the Item IDs that correspond to whole blood potassium level and serum potassium level. Similar to blood potassium, we extracted blood calcium and sodium levels using Item IDs based on Table II. Lastly, we exclude all the potassium measurements that were taken more than 3 hours compared to the time of ECG acquisition. Out of more than 800,000 ECG measurements in the MIMIC-IV-ECG dataset, only 169,287 of the records matched potassium measurements. This illustrates the need for a pretraining module where all the labeled and unlabeled data can be used for providing a regularized embedding space that can be finetuned for the downstream task with a minimal amount of labeled data.

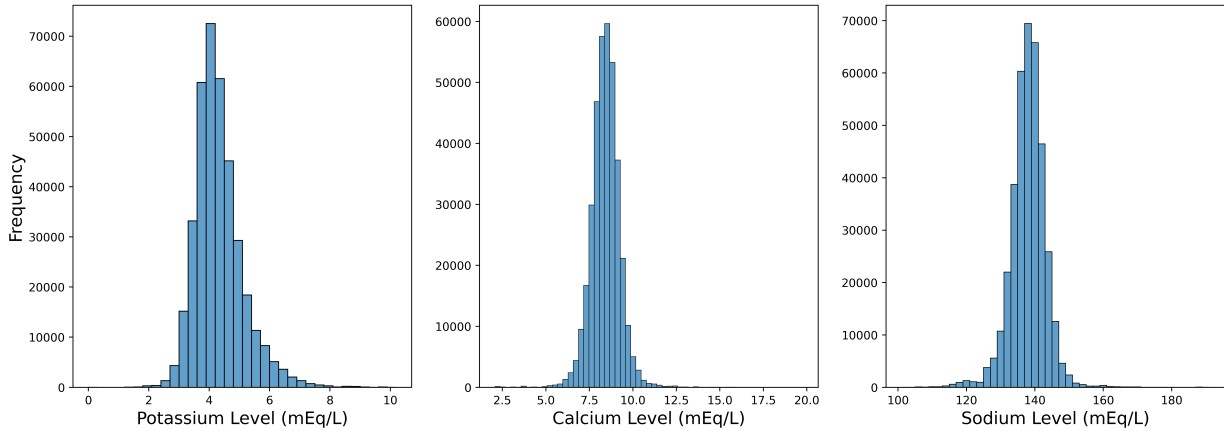

Fig. 2. Data distribution of potassium, calcium, and sodium measurements in the matched MIMIC-IV ECG dataset.

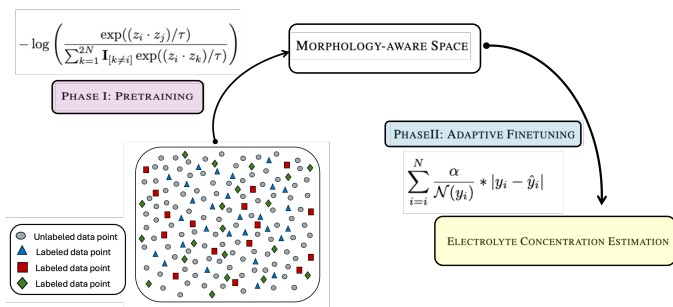

Fig. 3. Two-phase training process - ECG lead II and one other available ECG lead are used for phase 1: pertaining stage to train the feature encoder based on ECG morphology using the unlabeled dataset. In phase 2 of training, the labeled data is used with an adaptive loss to train the regressor for each electrolyte type.

## V. RESULTS

We used ResNet50 as the encoder backbone which comprised of 50 weighted convolutional layers to extract spatial ECG features. For the projector network, we used a multi-layer perceptron with 1024 dimensions. The model was pretrained using the backbone encoder and the projector network. For the finetuning process, the projector network was removed and the encoder followed by two fully connected layers to predict the outcome. The loss function for the pretraining stage is shown in Equation 2. The learning rate in this work is 0.01 coupled with a learning rate scheduler and a batch size of 256. We also applied the activation function ReLU for the projector layers. For the training, validation, and testing stages, we randomly selected 80% data for training and validation, and 20% for testing. The initial learning rates of 0.1, 0.01, and 0.001 were tuned using the validation dataset. Furthermore, the temperature variable used in this work is 0.5. This work is implemented in Python 3.9.15 and PyTorch 1.11.0 and the training was performed using 1 NVIDIA RTX A5000 GPU.

### A. Baseline Models

The neural network baselines incorporate the 10-layer CNN from [28], [30] and the ECG12Net from [29]. We constructed the unique, DenseNet-based ECG lead blocks of ECG12Net as per the author's specifications, utilizing 1-dimensional CNNs, max pooling, and batch normalization in PyTorch [29], [40] [1]. To streamline the process of estimating electrolyte concentration, we modified the final layers of both baselines for the training and validation of this task. For example, for the detection of dyskalemia, we transformed the predicted potassium concentration into tri-class labels, which allowed us to report performance.

We also adopted non-neural network baselines, XGBoost, and SVM from [26], by manually extracting morphologies (such as the QRS complex, T wave, etc.) from the same ECG waveform provided in MIMIC-IV-ECG data. We implemented the classification and regression models separately. To evaluate the effectiveness of the combination of the adaptive loss function and the pertaining step, we present multiple ablation studies as shown in Table III.

### B. Prediction tasks

We tested the model against the baselines in 2 different settings for three electrolyte types:

- Regressing the blood potassium, calcium, and sodium level
- Classifying three ranges of blood electrolyte level. For example, hypokalemia, normal, and hyperkalemia

In this work, we evaluated the classification results using Macro AUROC and Accuracy metrics. Given the presence of imbalanced classes, macro averaging was utilized to ensure that minority classes are represented equally as they hold equal importance in the evaluation. Having access to a predicted value of electrolyte level will enable a more granular prediction where the shift in this biomarker can be monitored for each patient. As shown in tables IV, V and VI, the proposed model resulted in lower mean absolute error and higher AUROC and accuracy.

---

[1]We implemented baselines to the best of our ability and per authors' specifications. We acknowledge that these baselines have reported higher metrics on the internal datasets. However, these results are our best effort at reimplementing and tuning these models that are not publicly available.

TABLE I
DETAILED DESCRIPTION OF STUDY COHORT

| | | Dataset | Train | Test |
|---|---|---|---|---|
| **Age(years), mean(SD)** | | 64.36(16.58) | 64.35(16.58) | 64.41(16.59) |
| **Gender, n(%)** | Female | 19116(44%) | 15134(44%) | 3982(44%) |
| | Male | 24379(56%) | 19391(56%) | 4989(55%) |
| **Race, n(%)** | White | 30043(69%) | 23799(69%) | 6245(69%) |
| | Black/African American | 4151(9%) | 3311(9%) | 840(9%) |
| | Hispanic/Latino | 1526(3%) | 1225(3%) | 301(3%) |
| | Asian | 1291(3%) | 1033(3%) | 258(3%) |
| | American Indian/Alaska Native | 70(0.1%) | 52(0.1%) | 18(0.2%) |
| | Native Hawaiian | 60(0.1%) | 45(0.1%) | 15(0.1%) |
| | South American | 44(0.1%) | 34(0.01%) | 10(0.01%) |
| | Multiple Race | 41(0.1%) | 30(0.09%) | 11(0.1%) |
| | Unknown | 6269(14%) | 4996(14%) | 1273(14%) |
| **BMI, mean(SD)** | | 28.9(7.3) | 28.9(7.3) | 28.8(7.3) |
| **Blood Pressure Measurement** | SBP, mean(SD) | 129.2(19.3) | 129.2(19.3) | 129(19.1) |
| | DBP, mean(SD) | 73.8(12.3) | 73.9(12.3) | 73.7(12.2) |
| **Potassium Measurement** | mean(SD) | 4.19(0.69) | 4.19(0.69) | 4.18(0.68) |
| | Hypokalemia | 16933 | 13445 | 3488 |
| | Hyperkalemia | 13631 | 10840 | 2791 |
| **Calcium Measurement** | mean(SD) | 8.47(0.85) | 8.47(0.84) | 8.46(0.85) |
| | Hypocalcemia | 35574 | 28294 | 7281 |
| | Hypercalcemia | 1432 | 1145 | 287 |
| **Sodium Measurement** | mean(SD) | 137.94(5.95) | 137.94(5.91) | 137.94(6.09) |
| | Hyponatremia | 17879 | 14218 | 3661 |
| | Hypernatremia | 7785 | 6218 | 1568 |

TABLE II
ITEM IDs AND DATA SOURCE INFORMATION IN MIMICIV DATASETS FOR DIFFERENT ELECTROLYTES

| Label | MIMICIV ItemID | Data Source | Category | Normal Range ($mmol/L$) |
|---|---|---|---|---|
| ZPotassium (serum) | 220640 | chartevents | Labs | 3.5 - 5 [37] |
| ZPotassium (whole blood) | 226535 | chartevents | Labs | 3.5 - 5 [37] |
| Potassium (serum) | 227442 | chartevents | Labs | 3.5 - 5 [37] |
| Potassium (whole blood) | 227464 | chartevents | Labs | 3.5 - 5 [37] |
| Calcium (serum) | 225625 | chartevents | Labs | 8.5 - 10.5 [38] |
| Calcium (whole blood) | 225667 | chartevents | Labs | 8.5 - 10.5 [38] |
| Sodium (serum) | 220645 | chartevents | Labs | 135 - 145 [39] |
| Sodium (whole blood) | 226534 | chartevents | Labs | 135 - 145 [39] |

## VI. LIMITATION AND FUTURE WORK

One main limitation of the proposed work is the lack of access to large amounts of labels, e.g., blood potassium levels from the clinical dataset for severe hypokalemia and hyperkalemia cases that could be matched with the ECG data. Although we introduced the adaptive loss function to mitigate this issue, a chief future direction will be to use other datasets and evaluate the server cases more comprehensively. Furthermore, in this work, we used a dataset where all 12 leads of ECG were available and selected 2 leads. In future work, we will employ other datasets with limited ECG leads. Lastly, as there are multiple ICU stays and ECG data for the same patient, in future work, we will analyze the change in the blood potassium level of one patient across time and the corresponding embedded value to understand the underlying association in more depth. In this work, we employed deep learning-based architectures for feature extraction from ECG biosignals. However, the interpretability of such models is crucial, particularly in clinical settings. Therefore, we will also incorporate interpretable features, such as attention maps, to allow for the visualization and interpretation of the extracted features, enabling a deeper investigation into the underlying characteristics of ECG signals across different conditions.

TABLE III

COMPARISON OF MODEL PERFORMANCES: 1: [28], [30], 2: [40], 3: [26], TRANSFORMER-BASED AND LSTM-BASED MODELS WHEN USING A COMBINATION OF THE PRETRAINING MODULE AND THE ADAPTIVE LOSS

| | Backbone | MAE | Macro AUROC | Accuracy |
|---|---|---|---|---|
| **Pre-trained** | 10-layer CNN [1] | 0.61 | 0.51 | 0.80 |
| | ECG12Net [2] | 0.57 | 0.51 | 0.81 |
| | Transformer | 0.53 | 0.51 | 0.80 |
| | LSTM | 0.60 | 0.53 | 0.81 |
| **Adaptive loss** | 10-layer CNN [1] | 0.60 | 0.56 | 0.80 |
| | ECG12Net [2] | 0.62 | 0.53 | 0.79 |
| | Transformer | 0.52 | 0.51 | 0.80 |
| | LSTM | 0.55 | 0.65 | 0.80 |
| **Both** | 10-layer CNN [1] | 0.55 | 0.51 | 0.80 |
| | ECG12Net [2] | 0.57 | 0.55 | 0.81 |
| | Transformer | 0.50 | 0.55 | 0.81 |
| | LSTM | 0.53 | 0.53 | 0.81 |
| | **Our work** | **0.49** | **0.69** | **0.82** |

TABLE IV

COMPARISON OF MODEL PERFORMANCES FOR POTASSIUM ESTIMATION

| | Potassium Concentration | Dyskalemia Detection | |
|---|---|---|---|
| Model | MAE | Macro AUROC | Balanced Accuracy |
| 10-layer CNN [28], [30] | 0.65 | 0.51 | 0.79 |
| ECG12Net [40] | 0.59 | 0.48 | 0.74 |
| XGBoost [26] | 0.76 | 0.53 | 0.79 |
| SVM [26] | 0.74 | 0.49 | 0.80 |
| Transformer | 0.55 | 0.51 | 0.80 |
| LSTM | 0.75 | 0.53 | 0.81 |
| **Our Work** | **0.49** | **0.69** | **0.82** |

TABLE V

COMPARISON OF MODEL PERFORMANCES FOR SODIUM ESTIMATION

| | Sodium Concentration | Dysnatremia Detection | |
|---|---|---|---|
| Model | MAE | Macro AUROC | Balanced Accuracy |
| 10-layer CNN [28], [30] | 7.65 | 0.53 | 0.73 |
| ECG12Net [40] | 6.93 | 0.49 | 0.78 |
| XGBoost [26] | 9.12 | 0.51 | 0.72 |
| SVM [26] | 9.41 | 0.5 | 0.79 |
| Transformer | 6.32 | 0.48 | 0.78 |
| LSTM | 8.82 | 0.51 | 0.76 |
| **Our Work** | **5.91** | **0.68** | **0.83** |

TABLE VI

COMPARISON OF MODEL PERFORMANCES FOR CALCIUM ESTIMATION

| | Calcium Concentration | Dyscalcemia Detection | |
|---|---|---|---|
| Model | MAE | Macro AUROC | Balanced Accuracy |
| 10-layer CNN [28], [30] | 0.79 | 0.49 | 0.78 |
| ECG12Net [40] | 0.71 | 0.48 | 0.8 |
| XGBoost [26] | 0.7 | 0.5 | 0.79 |
| SVM [26] | 0.81 | 0.52 | 0.78 |
| Transformer | 0.69 | 0.58 | 0.8 |
| LSTM | 0.75 | 0.53 | 0.81 |
| **Our Work** | **0.53** | **0.71** | **0.83** |

## VII. CONCLUSION

In this paper, we proposed a two-stage model for predicting electrolyte levels and classifying the hypo, normo and hyper ranges in blood potassium, blood calcium, and blood sodium levels. In this work, we used 10-sec diagnostic ECG data from MIMIC-IV dataset. Additionally, the ECG data was matched with MIMIC-IV clinical data to obtain the labels. For the first stage of the training, we utilized a contrastive pertaining framework for ECG data without the need for data augmentation where the embedding space is regularized by the ECG morphology. In the second stage, the morphology-aware embedding space was fine-tuned by labeled data for the biomarker estimation. As there is limited data from the severe abnormal ranges, we incorporated the inverse frequency as a weight augmentation method to ensure that the model is penalized more for the errors in severe abnormal ranges. This framework allowed leveraging a substantial amount of unlabeled data in the pretraining phase, presenting the capability to dynamically fine-tune and estimate biomarkers, ultimately yielding superior performance. The proposed framework will enable us to use ECG data for electrolyte monitoring which provides benefits in early detection, preventive action, and overall health management. By providing real-time data we aim to enable proactive monitoring, alerting patients and healthcare providers to deviations in electrolyte levels before symptoms manifest, allowing for timely intervention. Additionally, the non-invasive data can be used with time-to-event models to enable recurrent event triggers to prevent severe abnormalities in electrolyte levels which can result in other health conditions such as renal failure.

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
