# OpenReview forum: "Non-invasive Electrolyte Estimation Using Multi-lead ECG data via Semi-supervised Contrastive Learning with an Adaptive Loss"
_IEEE.org/EMBS/BHI/2024/Conference — IEEE BHI'24_

### Official Review · Reviewer_TZSk · 2024-08-08
**Great potential, unclear methodological approach**

**Overall Rating:** 6
**Confidence:** 3

**Other Quality Metrics:**

The clarity of the writing is generally clear, providing sufficient information to fully replicate the proposed work.

The study's clinical significance could be considerable, especially given the growing importance of non-invasive real-time monitoring in both clinical practice and personal wearable devices. The opportunistic approach of detecting electrolyte deficiencies solely through ECG data, without the need for additional hardware, could pave the way for more comprehensive monitoring solutions for patients, athletes, and the general population. This method holds the potential to significantly enhance health management by providing continuous, cost-effective monitoring of electrolyte levels.

The methodological novelty of this work makes it suitable for publication at the BHI conference. However, there are a few concerns regarding the mathematical assumptions underlying the study that require clarification or adjustment before publication. Addressing these issues will strengthen the reliability and credibility of the findings.

The results are somewhat challenging to interpret within the context of evaluating electrolyte deficiencies from opportunistic biosignals, such as ECG. The authors compare different model backbones in terms of performance but do not mention how their approach compares to other non-invasive biomarker estimation methods. This omission makes it difficult to gauge the relative effectiveness of their method. Furthermore, the reviewer is unable to assess the criticality of the accuracy achieved by the authors. For instance, is an 82% accuracy rate sufficient for this scenario, or could failing to detect 18% of cases lead to serious pathological conditions? Providing a discussion on the clinical implications of the achieved accuracy and its impact on patient outcomes would be beneficial.

**Questions For The Authors:**

See Weaknesses section

**Strengths:**

The idea of assessing the level of electrolytes using the ECG signal is quite powerful and could lead to a massive improvement of non-invasive patient monitoring systems.
The analysis of the limitations is well detailed and thorough.

**Summary Of The Paper:**

This work addresses the challenges of predicting electrolyte levels from ECGs using machine learning models, particularly focusing on the scarcity of labeled data and the Gaussian distribution of electrolyte values in datasets, which leads to fewer extreme values crucial for prediction. The authors propose a solution using a pretrained ECG encoder trained on large, unlabeled datasets to improve generalization and an adaptive loss function that regularizes backpropagation inversely to label frequency. This approach results in significant improvements in mean absolute error (MAE) for blood potassium, calcium, and sodium level regression tasks (reductions of 0.1, 0.16, and 0.41, respectively), and more than a 0.15 increase in the area under the receiver operating characteristic curve (AUROC) for abnormal electrolyte classification tasks compared to state-of-the-art models.

**Weaknesses:**

Although the work is remarkable, the results are somewhat difficult to read given the complex sets of assumptions used in this work.

Problems
- In subsection C of section III, the authors use a Shapiro-Wilk test to verify if the samples come from a normally distributed population. However, since the result is below 0.0001, the population is not normally distributed, contrary to what the authors claim. In fact, in the Shapiro-Wilk test, a low p-value (typically below 0.05) indicates that you can reject the null hypothesis (a sample comes from a normally distributed population), suggesting that the data do not follow a normal distribution. If your p-value is lower than 0.0001, it means the data likely do not follow a normal distribution, contradicting the assumption that the biomarker value follows a normal distribution. This somehow invalid all the following approaches. Please, consider examining the data using other statistical methods or transformations (e.g., Box-Cox transformation) that might normalize the distribution if needed. Alternatively, use a data-driven approach to determine sample weights, potentially relying on empirical frequencies rather than assumed distributions. Also try to rethink your assumptions. If normality is a requirement for your analysis, consider why the data might deviate from normality and address it through preprocessing or different modeling approaches.
- The description of the Adaptive MAE is quite confusing. What does this acronym stand for? Has it been used in other similar works? What are N, i, N, y_i and y ̂_i? If y and α are tunable coefficients, what are the parameters used in this work?

Comments and typo
- Define the acronym ICU in Section III subsection A.
- Explain better what is the “Anchor lead II data point” mentioned at the end of page 3 and add a reference if possible.
- Substitute the header “Race” in table I with “Ethnicity”. Apart from ethical considerations, "race" is not a scientifically rigorous way to categorize human populations based on genetic differences. While “Ethnicity” can more accurately reflect cultural, linguistic, geographical, and historical backgrounds without implying inaccurate biological distinctions. Moreover, some of the descriptors, e.g. “White” and “Asian”, are quite wide in terms of genetic variations. Race instead should be applied thoughtfully and in a manner that acknowledges its complexities.
- Please provide a detailed explanation of how you calculate the AUROC and Accuracy in your work. These metrics can vary slightly depending on the data type and the algorithmic approach used, so it's important to describe mathematically how these values are obtained. Alternatively, you can reference a work that explains these calculations.

---

### Official Review · Reviewer_tvYt · 2024-08-08
**Non-invasive Electrolyte Estimation Using Multi-lead ECG data via Semi-supervised Contrastive Learning with an Adaptive Loss**

**Overall Rating:** 7
**Confidence:** 5

**Other Quality Metrics:**

(a) Excellent
(b) Good
(c) Excellent
(d) Great

**Questions For The Authors:**

Can you check and compare your method across different ECG conditions?
What are underlying characteristics representing ECG which are minimized across two ECG leads?

**Strengths:**

Novel approach in which researchers involve self-supervised contrastive learning to train ECG features
Excellent demonstration of results compared to State of the art
Can be used to estimate other physiological signs utilizing ECG

**Summary Of The Paper:**

This paper is concerned with non-invasive electrolyte estimation via ECG. The paper suggest novel approach semi-supervised contrasting learning architecture to train 2-lead ECG feature by minimizing the difference in underlying characteristics.

**Weaknesses:**

Could have involved explaining what type of features have been extracted in terms of underlying characteristics for clinical explain ability and significance

---

### Official Review · Reviewer_fX8T · 2024-08-11

**Overall Rating:** 7
**Confidence:** 4

**Other Quality Metrics:**

(a) Clarity of writing: good

(b) Clinical Significance: fair

(c) Methodological Novelty: fair

(d) Experiments and Results: excellent

**Questions For The Authors:**

N/A

**Strengths:**

1. The paper is well-written and conveys the idea clearly.
2. The general idea of using SimCLR to train the model, including the definition of positive and negative pairs, makes sense.
3. The extensive experiments show the promising performance of the proposed method.

**Summary Of The Paper:**

The paper presents a method to estimate electrolyte levels using ECG data, using semi-supervised contrastive learning with an adaptive loss function. It addresses challenges in training machine learning models due to label scarcity and data distribution issues. By leveraging large unlabeled datasets and a novel loss approach, the model achieves improved accuracy in predicting electrolyte levels, showing potential for enhancing patient monitoring with non-invasive methods.

**Weaknesses:**

*The real-world applicability of this work is not very solid.* The method addresses a diagnosis task, which predicts electrolyte levels using the current ECG signal. My main question is what can you do with this diagnosis prediction? At the time the model makes the prediction, the patient might just have abnormal electrolyte concentration, and the symptoms (for example muscle spasms) already happened. If the patient was driving at this exact time, the accident might also happened as well. Your prediction, although accurate, cannot help prevent the abnormal electrolyte concentration and accident.

---

### Decision · Program_Chairs · 2024-09-22

Accept